# LSTrAP-Cloud: A User-Friendly Cloud Computing Pipeline to Infer Coexpression Networks

**DOI:** 10.3390/genes11040428

**Published:** 2020-04-16

**Authors:** Qiao Wen Tan, William Goh, Marek Mutwil

**Affiliations:** School of Biological Sciences, Nanyang Technological University, 60 Nanyang Drive, Singapore 637551, Singapore; qiaowen001@e.ntu.edu.sg (Q.W.T.); will0046@e.ntu.edu.sg (W.G.)

**Keywords:** cloud, RNA, sequencing, coexpression, metabolism

## Abstract

As genomes become more and more available, gene function prediction presents itself as one of the major hurdles in our quest to extract meaningful information on the biological processes genes participate in. In order to facilitate gene function prediction, we show how our user-friendly pipeline, the Large-Scale Transcriptomic Analysis Pipeline in Cloud (LSTrAP-Cloud), can be useful in helping biologists make a shortlist of genes involved in a biological process that they might be interested in, by using a single gene of interest as bait. The LSTrAP-Cloud is based on Google Colaboratory, and provides user-friendly tools that process quality-control RNA sequencing data streamed from the European Nucleotide Archive. The LSTRAP-Cloud outputs a gene coexpression network that can be used to identify functionally related genes for any organism with a sequenced genome and publicly available RNA sequencing data. Here, we used the biosynthesis pathway of *Nicotiana tabacum* as a case study to demonstrate how enzymes, transporters, and transcription factors involved in the synthesis, transport, and regulation of nicotine can be identified using our pipeline.

## 1. Introduction

Genome sequencing and assembly are becoming more accessible in terms of cost and computational resources required due to the advances in technology and algorithms [1]. However, elucidating gene function is necessary to extract meaningful knowledge from genomes. Despite extensive efforts over the decades, only 12% of genes have been characterised in the most studied model plant Arabidopsis thaliana [2]. This is because gene characterization is a time- and labor-intensive process hindered by various obstacles such as the lethality of mutants involving essential genes, or conversely, no observable mutant phenotype due to functional redundancy caused by large gene families [3,4,5,6,7]. Evidently, unguided experimental characterization of all genes is not feasible and, consequently, computational gene function prediction studies are conducted to meet this challenge (reviewed in Rhee and Mutwil [3]). To this end, newly sequenced genomes are mostly annotated using sequence similarity approaches, which annotate novel genes based on the sequence similarity to characterized genes. While sequence similarity analysis gives a quick overview of gene functions in a new genome, it has its limitations as genes can (i) have multiple functions, (ii) sub- or neofunctionalise via gene duplications, or (iii) have no sequence similarity to characterized genes. Most importantly, sequence similarity approaches often cannot reveal which genes work together in a novel biological process, for example, a specialized metabolic pathway. Clearly, classical approaches to gene function prediction are powerful but require other methods to complement them [3].

In order to associate genes to pathways, one has to consider the expression of genes at the level of organs, tissues, and even cells. With the increasing availability of RNA sequencing (RNA-seq) data, it is now possible to study genes from the perspective of their expression [3,8]. Genes that have similar expression profiles across different organs, developmental stages, time of the day, and biotic and abiotic stress conditions tend to be functionally related [3,6,8,9,10,11]. These transcriptionally coordinated (coexpressed) genes can be revealed by analyzing transcriptomic data stemming from microarrays or RNA-seq data. In turn, coexpressed genes can be represented as nodes connected by edges (links) in coexpression networks, which can be mined for groups of highly connected genes (modules, clusters) that are likely to be involved in the same biological process. Coexpression networks have become a popular tool to elucidate the function of genes, their related biological processes, and gene regulatory landscapes. Genes involved in a wide range of processes, including cellular processes [12,13,14], transcriptional regulation [15], physiological responses to the environment and stress [9,16], and plant viability and the biosynthesis of metabolites [17,18,19,20,21] have been elucidated using coexpression analyses.

The amount of gene expression data has grown tremendously over the decade, with a more than 1000-fold increase in nucleotide bases on NCBI Sequence Read Archive (SRA), from 11TB to 12 PB at the start of 2010 and 2020, respectively. Analysing this data would have been unthinkable a decade ago, due to limitations in software used to estimate gene expression from RNA-seq data. However, drastic improvements in software used to estimate gene expression from RNA-seq data, such as kallisto [22] and salmon [23], have made this task possible within a reasonable time on a typical desktop or even a Raspberry Pi-like miniature computer [24]. Furthermore, multiple user-friendly pipelines are an invaluable resource both for experts and nonbioinformaticians, to which pipelines such as UTAP [25], LSTrAP-Lite [24], and LSTrAP [26] are made publicly available. However, all these resources typically require complex installation or a linux environment.

The introduction of cloud computing has provided alternatives to how data can be managed, stored, and processed. Google colaboratory (colab), a Jupyter notebook environment, was launched in 2017, and it allows users to write and execute python code on Google’s cloud servers through their browser (https://colab.research.google.com/). The use of the colab platform for RNA-seq analysis was demonstrated by Melsted et al., 2019 [27], who implemented the workflow of preprocessing single cell transcriptomic data. The choice of colab greatly improves user friendliness with the clean layout of jupyter notebooks, a graphical interface that is more friendly for biologists than the typical linux terminal, and problems associated with installation in different local environments. Collaboration between scientists is also easily established as notebooks can be saved to Google Drive and Github and deployed on a new computer within a minute. Most importantly, colab provides the computing power to perform RNA-seq analysis for free, allowing users to run computationally heavy calculations using any computer, tablet or cell phone.

These advances and resources have prompted us to showcase the use of colab as a seamless and user-friendly interface for large-scale transcriptomic analysis. We illustrate this by using RNA-seq data of the model plant *Nicotiana tabacum* to dissect the biosynthesis of nicotine by coexpression network analysis. The presented pipeline, the Large-Scale Transcriptome Analysis Pipeline in Cloud (LSTrAP-Cloud), is available from https://github.com/tqiaowen/LSTrAP-Cloud, and can be easily applied to other pathways and organisms.

## 2. Materials and Methods

### 2.1. Streaming RNA Sequencing Data

The pipeline (Figure 1) was implemented on Google Colaboratory and consists of two jupyter notebooks that are available on https://github.com/tqiaowen/LSTrAP-Cloud. RNA sequencing experiments (Appendix A) were streamed as fastq files from the European Nucleotide Archive (ENA) [28] using curl v7.58.0 with the parameters “-L -r 0-1,000,000,000 -m 600 –speed-limit 1,000,000 –speed-time 30”, which allows curl to retrieve the first billion bytes (953 megabytes) of the fastq file over a maximum duration of 600 s. The streaming is also aborted if the download speed drops below 1 million bytes per second (1 Mb/s) for 30 s. Files ending with “_1.fastq.gz” and “.fastq.gz” were downloaded for paired and single library layouts, respectively. The streamed data from curl was piped to kallisto quant v0.46.0 [22] with the parameters “–single -l 200 -s 20 -t 2” (single end experiment with read length of 200 bp, standard deviation of 20 and to run with two threads) and mapped against the kallisto index of coding sequences (CDS) of *N. tabacum* [29]. CDS Nitab-v4.5_cDNA_Edwards2017.fasta was obtained from SolGenomics [30] and used to generate kallisto index with default parameters. A total of 1049 out of 1060 experiments were processed successfully. Seven files were not found, and four files had unacceptable download speed among the files that were not processed successfully (Appendix A).

### 2.2. Constructing Coexpression Networks

Coexpression network of the gene of interest, *Nitab4.5_0000884g0010.1*, was obtained by including a maximum of 50 other genes with a Pearson Correlation Coefficient of at least 0.7 against the gene of interest. The coexpression network of *Nitab4.5_0000884g0010.1* was visualised on colab using Cytoscape.js v3.9.4 [31]. The shapes and colours of the genes were assigned according to the major Mapman bin classification obtained from Mercator4 v2.0 [32] with the *N. tabacum* Nitab-v4.5_cDNA_Edwards2017.fasta CDS (Appendix A). The coexpression neighbourhood of the gene of interest is also displayed at the end of the colab notebook.

To annotate and summarise the coexpression network of *Nitab4.5_0000884g0010.1*, the JSON file of the network was downloaded from colab and modified in Cytoscape desktop v3.7.1 (Appendix A). For brevity, only transporters, transcription factors and genes involved in nicotine biosynthesis are shown, but the network containing all 50 genes is available (Appendix A).

### 2.3. Identification of Genes Involved in Nicotine Biosynthesis

Nucleotide sequences of genes that reported to be involved in nicotine biosynthesis and transport (ODC1: AB031066; ODC2: AF233849; PMT: D28506; MPO1: AB289456; MPO2: AB289457; AO: XM_016633697, XR_001648132; QS: XM_016642986, XM_016638757; QPT1: AJ748262; QPT2: AJ748263; A622: D28505; BBLa: AB604219; BBLb: AM851017; BBLc: AB604220; BBLd: AB604221; MATE1: AB286963; MATE2: AB286962 and NUP1: GU174267) [33] were retrieved from NCBI. The gene IDs of these genes in the tobacco genome version Nitab-v4.5_cDNA_Edwards2017.fasta were identified through blast v2.6.0+ against the *N. tabacum* CDS. The function of the genes found in the network was further annotated using results from blast and Mercator (Appendix A).

### 2.4. Relative Expression of Nicotine Biosynthesis Genes in 5 Major Organs

To obtain the relative expression of the genes shown in the coexpression network of *Nitab4.5_0000884g0010.1*, annotation of experiments was retrieved from NCBI SRA run selector (Appendix A). Only wild-type and untreated experiments indicating leaf, flower, root, shoot and stem were selected for the analysis (Appendix A). The median expression value of a gene in an organ was normalised with the highest median expression value of the gene across all organs.

## 3. Results

### 3.1. Implementation of Gene Coexpression Pipeline on Google Colaboratory

The improvement in transcript estimation algorithms has greatly reduced the amount of time and resources required to estimate gene expression from RNA-sequencing data. Previously, we have demonstrated with the LSTrAP-Lite pipeline that analysis of large-scale transcriptomic data was possible on a small computer costing less than 50 USD [24]. However, user friendliness of the LSTrAP-Lite pipeline was still limited, as it runs in the linux terminal and Advanced RISC Machine (ARM) CPU architecture, which is not user-friendly to most biologists and not compatible with most software, respectively.

Here, we implemented a large-scale transcriptomic analysis pipeline on Google Colaboratory, a free cloud computing platform that allows multiple users to easily share and deploy python code in a jupyter notebook environment. The pipeline (Figure 1), Large-Scale Transcriptome Analysis Pipeline in Cloud (LSTrAP-Cloud), takes the CDS of the organism of interest, and streams the list of RNA-seq experiments specified by the user from ENA. After all files have been streamed, quality statistics from the download report are displayed on the notebook and summarised in plots. The plots allow the user to set an appropriate cutoff for experiments to be included in downstream analyses. Lastly, the pipeline generates and displays a coexpression network of the gene of interest, which can be downloaded as a PNG or JSON file. All outputs from the notebook are saved in the Google Drive account of the user, which is needed to run the notebook. We provide a user’s manual (Appendix A) and SRA experiment list (Appendix A), allowing the readers to replicate this analysis.

The performance of the LSTrAP-Cloud was evaluated on the gene expression data of *N. tabacum*, a commercially important crop for the production of tobacco and an important model used in plant research. The first billion (953 megabytes) bytes of all publicly available RNA sequencing experiments of *N. tabacum* were streamed from ENA and processed by kallisto. An average processing speed of 4 Mb/s was achieved (Figure 2A), and an average of 17 million reads was pseudoaligned per 1 Gb of data (Figure 2B). Overall, 796 Gb of data were streamed in 59 h (Figure 2C), which is equivalent to 4 min per Gb. Quality statistics revealed that most experiments had more than 10^7^ reads processed by kallisto, where 60–80% of the reads were pseudoaligned to the CDS (Figure 2D), and 50–70% of the CDSs had a nonzero expression (Figure 2E). Under the assumption that most samples are of good quality, we chose RNA-seq experiments that had at least one million reads pseudoaligned to the *N. tabacum* CDS, 40% of streamed reads mapped to the CDS, and at least 40% of genes with nonzero (Figure 2E). Nine hundred and sixty-two experiments passed these conditions and were compiled into an TPM expression matrix, where genes were arranged in rows and experiments in columns (Appendix A).

### 3.2. Investigating the Nicotine Biosynthesis Coexpression Network

Nicotine is a toxic alkaloid produced by plants in the Solanaceae family to deter herbivores and a potent addictive substance. The synthesis of nicotine occurs in the roots and involves two precursors, a pyrrolidine (N-methyl-∆1-pyrrolinium cation) and a pyridine (nicotinic acid) ring derived by a series of reactions from ornithine and aspartate, respectively ([34], Figure 3A). The rings are then combined by the enzymes A622 (isoflavone reductase) and berberine bridge enzyme-like (BBL) to form nicotine. After synthesis in the roots, nicotine is sequestered out of the roots by multidrug and toxic compound extrusion (MATE) family transporters [35] and accumulated in organs that are highly prone to attack by herbivores, such as leaves [36].

Coexpression networks have been shown to be useful in the identification of enzymes, transcription factors, and probable transporters of plant metabolites [11,20,24,37]. To demonstrate that this is also true in the case of nicotine biosynthesis, we retrieved the top 50 genes coexpressed with A622 (*Nitab4.5_0000884g0010.1*) (Figure 3B). The network revealed enzymes that are known to be involved in nicotine biosynthesis (aspartate oxidase (AO), quinolate synthase (QS), quinolinate phosphoribosyltransferase 2 (QPT2), ornithine decarboxylase 2 (ODC2), putrescine N-methytransferase (PMT), N-methylputrescine oxidase (MPO), and BBL) and transcription factors (APETALA 2/ethylene responsive factor (AP2/ERF) and basic helix-loop-helix (bHLH)], which regulate nicotine biosynthesis [38]. In addition to the expected MATE2 transporters, other transporters such as organic cation transporter (OCT), nicotinate transporter (NiaP), Usually Multiple Amino Acids Move In and Out Transporter (UmamiT), and purine uptake permease (PUP) are also observed. To conclude, from the coexpression network of A622, we observed that more than half of the genes in the network (27 out of 50) are involved in nicotine biosynthesis. The remaining genes found in this coexpression network are excellent candidates for further functional analysis of their involvement in nicotine biosynthesis.

### 3.3. Expression Analysis of Genes Related to Nicotine Biosynthesis in Flower, Leaf, Root, Shoot, and Stem

It is well established that the synthesis of nicotine occurs in the root, and numerous analyses have shown that secondary metabolism is under strong transcriptional control [7,24,39,40]. Hence, the expression of nicotine biosynthesis genes should be either specific or highly expressed in the roots. To confirm this, we calculated the median gene expression value of the genes found in the coexpression network (Figure 3) in roots, leaves, flowers, shoots, and stems (Figure 4). As expected, all enzymes directly involved in nicotine biosynthesis are most expressed in roots and lowly expressed in other organs. Thus, we can conclude that the genes identified in the network of A622 have root specific expression in *N. tabacum* and are very likely to be involved in nicotine biosynthesis.

## 4. Discussion

As we sit on an expanding trove of data today, there is an immense amount of knowledge to be uncovered with the improvement in gene annotation and characterization. Classical genomic approaches have allowed us to rapidly annotate genomes in silico based on sequence similarity to existing sequences. This approach has its limitations but can be greatly improved when the spatial and temporal expression of genes is taken into account.

In this study, we leveraged on the benefits of cloud computing and the user-friendliness of the Jupyter notebooks to implement a large-scale transcriptomic analysis pipeline, LSTrAP-Cloud (Figure 1). Using *N. tabacum* as an example (Figure 2), we showed that coexpression networks not only identified the enzymes involved in the metabolism of nicotine but also regulators and transporters that are found up- and down-stream of nicotine biosynthesis (Figure 3 and Figure 4). The example of nicotine biosynthesis demonstrates that coexpression networks analysis is a valuable addition to sequence similarity-based approaches, as it can infer modules of functionally related genes.

While the field of bioinformatics is advancing rapidly, it is important that biologists are also empowered with the tools and predictions available to bioinformaticians as this can greatly shorten the amount of time required for gene characterization through the identification of potential targets. The future of gene function prediction, however, will require a new generation of biologist equipped to tackle both wet and dry lab as sequencing data become available at a faster and larger rate [41].

## Figures and Tables

**Figure 1 genes-11-00428-f001:**
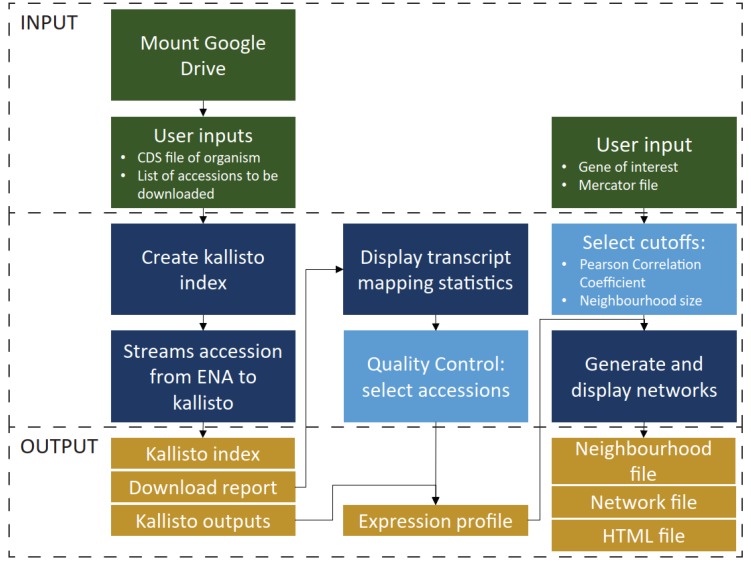
Schematic of the Large-Scale Transcriptomic Analysis Pipeline in Cloud (LSTrAP-Cloud) pipeline used for streaming and mapping of RNA sequencing data, and for the generation of coexpression network based on the gene of interest.

**Figure 2 genes-11-00428-f002:**
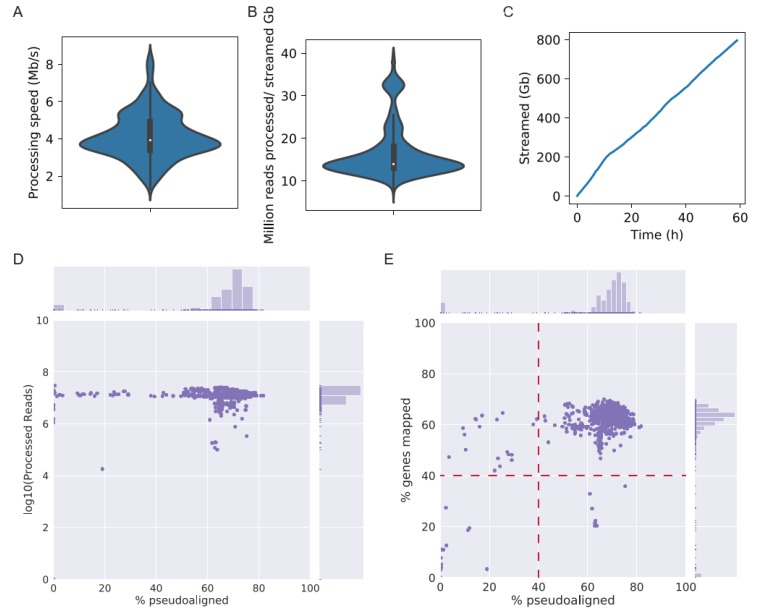
Summary of tobacco experiments streamed and processed by LSTrAP-Cloud. (**A**) The violin plot shows the speed of simultaneous streaming and processing of RNA-seq files by kallisto. (**B**) Amount of reads processed per gigabyte streamed. (**C**) Cumulative size (*y*-axis) of files streamed over time (*x*-axis). (**D**) The scatterplot shows the percentage of reads pseudoaligned to the coding sequences (CDS) (*x*-axis) against the total number of streamed reads (*y*-axis) for each experiment, which are represented as dots on the plot. (**E**) The percentage of reads pseudoaligned to the CDS (*x*-axis) versus the percentage of genes with nonzero Transcripts Per Kilobase Million (TPM) values (*y*-axis) for each experiment, which are represented as dots on the plot. Red lines indicate cutoffs that were used to the select experiments for downstream analyses.

**Figure 3 genes-11-00428-f003:**
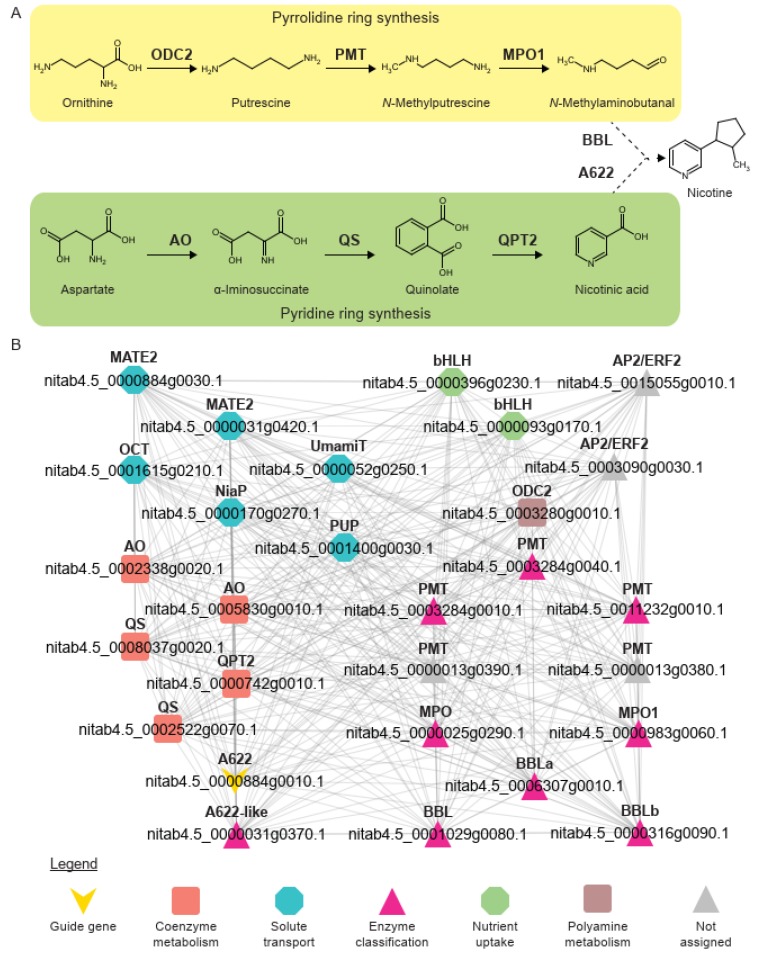
Nicotine biosynthesis. (**A**) Schematic of the nicotine biosynthesis pathway. Abbreviations for the enzymes are ODC2: ornithine decarboxylase 2, PMT: putrescine N-methyltransferase, MPO1: N-methylputrescine oxidase 1, BBL: berberine bridge enzyme-like proteins, AO: L-aspartate oxidase, QS: quinolinate synthase, QPT2: quinolinate phosphoribosyltransferase 2, A622: isoflavone-like oxidoreductase. (**B**) Coexpression network of A622 (*Nitab4.5_0000884g0010.1*). Abbreviations are AP2/ERF: APETALA 2/ethylene responsive factor, bHLH: basic helix-loop-helix, MATE2: multiantimicrobial extrusion family protein 2, OCT: organic cation transporter, NiaP: nicotinate transporter, PUP: purine uptake permease, and UmamiT: Usually Multiple Amino Acids Move In and Out Transporters. For brevity, only homologs of genes involved in nicotine biosynthesis as described in (**A**), transporters, and transcription factors are shown.

**Figure 4 genes-11-00428-f004:**
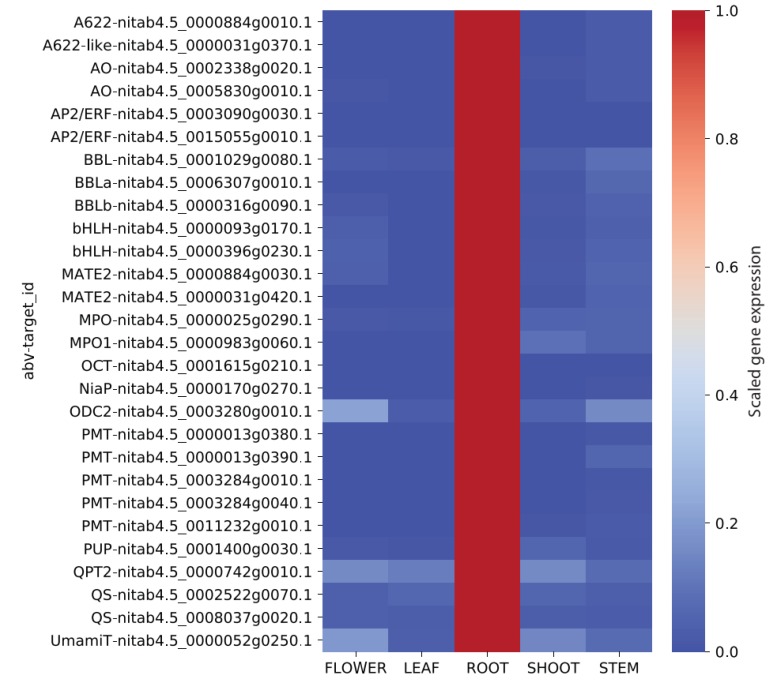
Relative expression of genes across major organs in *Nicotiana tabacum*. The expression values for each gene were scaled by dividing each row by the maximum value found in the row.

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
