# Peer review of "LSTrAP-Cloud: A User-Friendly Cloud Computing Pipeline to Infer Coexpression Networks"

_genes, 2020, doi:10.3390/genes11040428_

Round 1

Reviewer 1 Report

The manuscript describes a pipeline for conducting co-expression network analysis of a large number of RNA-Seq experiments directly from ESA in the cloud. This seems like a very useful approach especially for biologist with little computational training. I only have a few comments:

Tittle: “to Infer Co-functional and Regulatory Networks”. I find this title promising way too much and not in line with what the authors describe throughout their manuscript. I strongly suggest you change to “to infer co-expression networks”.

Abstract: “helping biologists make a shortlist of genes that they might be interested in”. Great, but reading the abstract I’m wondering: What is the question these biologist have and what do they input? I think the abstract should clearly state what the user inputs and what type of question this pipline solves.

Small issues:

p. 1: “other methods to complement it [3]” it -> them?

p. 2: “heavy calculations on any computer, tablet or cell phone”. Replace “on” with “via” or “using”?

p. 2: “which allows curl to be redirected to updated address, retrieve first 1 billion bytes”. Something needs to be rewritten in this sentence. I do not follow.

p. 3: “To generate Figure 3, The” small T.

p. 3: “user friendliness of _the_ LSTrAP-Lite pipeline”

p. 4: Fig 1 have diamonds with question marks in them several places in the text.

p. 4: “in a Google Drive account of the user” a -> the. Also this point is made again one sentence down.

p. 4: “The first billion (109) bytes”. What does 109 mean here?

p. 4: “most experiments had more than 107 reads processed” 107 reads is nothing? I don’t understand what the authors are trying to say here.

p. 4: “50-70% of the CDS had a non-zero expression” CDS -> CDSs

Fig 2: Panel B has a visible box around it. The D-label is clearly in the wrong place.

Fig 2 text: D: “the percentage of reads pseudoaligned to the CDS (x-axis) against the total number of streamed reads (y-axis)”. Shouldn’t you add “for each sample/experiment” or something? What is the individual dots in the plot? Same for E.

p. 5: “then combined to by the enzymes”. Remove “to”?

p. 7: “Thus, we can conclude that the genes identified in the network of A622 are specifically expressed in the roots of N. tabacum are very likely to be involved in nicotine biosynthesis.”. Rewrite. “and” in front of the second “are”?

Reviewer 2 Report

In this manuscript, the authors illustrate how to use Google colaboratory (colab) for large scale transcriptomics analysis using the Large-Scale Transcriptome Analysis Pipeline in Cloud (LSTrAP-Cloud). Colab is a free Jupyter notebook environment that requires no setup and that works entirely in the cloud where users can write and execute python code on the cloud servers of Google through their browser and save and share their analyses for free. To the aim of the manuscript, the authors use RNA-seq data of the model plant Nicotiana tabacum to dissect the biosynthesis of nicotine by co-expression network analysis. LSTrAP-Cloud is available from https://github.com/tqiaowen/LSTrAP-Cloud.

The manuscript is well written.  The authors provide all the necessary information and material to reproduce the analysis described and the example provided can be useful to other scientists in the field as a guideline for applications to different pathways and organisms.

Minor comments

In Figure 1 replace question marks with “i”.

Also, Figure 2 should be checked. There is no correspondence of the letters to indicate the different figure boxes.

Supplementary tables are all in one file that is named SuppTable2, that seems that only table 2 is provided.  This induces in confusion, it should be named in a different way.

In the main body of the manuscript, references should always be cited by using the same notation. To use the numerical one and then the name of authors make it difficult to follow.
